# CAUSAL SCREENING TO INTERPRET GRAPH NEURAL NETWORKS

## ABSTRACT

With the growing success of graph neural networks (GNNs), the explainability of GNN is attracting considerable attention. However, current works on feature attribution, which frame explanation generation as attributing a prediction to the graph features, mostly focus on the statistical interpretability. They may struggle to distinguish causal and noncausal effects of features, and quantify redundancy among features, thus resulting in unsatisfactory explanations. In this work, we focus on the causal interpretability in GNNs and propose a method, *Causal Screening*, from the perspective of cause-effect. It incrementally selects a graph feature (*i.e.,* edge) with large causal attribution, which is formulated as the individual causal effect on the model outcome. As a model-agnostic tool, Causal Screening can be used to generate faithful and concise explanations for any GNN model. Further, by conducting extensive experiments on three graph classification datasets, we observe that Causal Screening achieves significant improvements over state-of-the-art approaches *w.r.t.* two quantitative metrics: predictive accuracy, contrastivity, and safely passes sanity checks.

## 1 INTRODUCTION

Graph neural networks (GNNs) (Gilmer et al., 2017; Hamilton et al., 2017; Velickovic et al., 2018; Dwivedi et al., 2020) have exhibited impressive performance in a wide range of tasks. Such a success comes from the powerful representation learning, which incorporates the graph structure with node and edge features in an end-to-end fashion. With the growing interest in GNNs, the explainability of GNN is attracting considerable attention. A prevalent technique is to offer post-hoc explanations via feature attribution. The attributions in GNNs are typically defined as the contributions of input features (*e.g.,* nodes and edges) to the model's outcome; thereafter, by selecting the most important features with top attributions, an explanatory subgraph is constructed to answer "Why this GNN model makes such predictions?". In this line, current works roughly fall into two categories: (1) decomposing the outcome prediction to graph structures via backpropagating the gradient-like signals (Pope et al., 2019; Baldassarre & Azizpour, 2019); and (2) approximating the decision boundary via structure perturbations (Huang et al., 2020) or structure masking (Ying et al., 2019).

However, these works mostly focus on the statistical interpretability (Pearl, 2018; Moraffah et al., 2020), which could fail to uncover the causation of model predictions reliably. The key reason is that, they approach the input-outcome relationships from an associational standpoint, without distinguishing between causal and noncausal effects. Using correlation as causation to interpret feature importance will result in unfaithful explanations, as shown in the running example.

**Running Example.** Consider the example in Figure 1, where SA (Baldassarre & Azizpour, 2019) and GNNExplainer (Ying et al., 2019) use gradients and masks as attributions respectively, to explain why the scene type of a scene graph is predicted as *Surfing* by APPNP (Klicpera et al., 2019). Two limitations in the statistical interpretability are: (1) Confounding association. The edges with large gradient (*e.g.,* (*shorts, on, man*)) or masks (*e.g.,* (*man, has, hand*)) are highly correlated with the prediction, rather than causing it (Moraffah et al., 2020). Such confounding associations distort the estimation of the causation (*e.g.,* (*standing, on, surfboard*)); (2) Redundancy. As the graph structure is highly entangled with GNNs, the gradient-like signals of edges are influenced, even scaled, by the connected edges. This makes redundant edges (*e.g.,* (*man, on ocean*) and (*man, riding, waves*)) involved in the top explanations, while forgoing other edges with unique information.

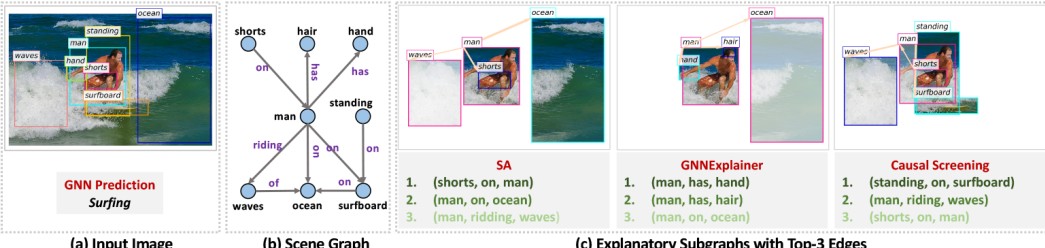

Figure 1: An example of explaining the scene graph classification. (a) An input image with bounding boxes; (b) The scene graph predicted as *Surfing*; (c) Explanations of SA, GNNExplainer, and our Causal Screening.

In this work, we focus on the causal interpretability (Pearl, 2018; Moraffah et al., 2020) in GNNs. From a cause-effect standpoint, we need to answer causality-related questions like "Was it a specific edge that caused the GNN's prediction?". Technically, explanatory subgraphs of interpretable features should account for two desirable characteristics: (1) Causality association. We need to identify important features, which may plausibly be causal determinants of the model outcome, away from these associated with the outcome due to confounding. As Figure 1 shows, (*standing, on, surfboard*) should be ranked at the top, instead of (*shorts, on, man*). (2) Briefness. We need brief explanations to avoid redundancy. For example, (*man, on, surfboard*) is informative but redundant since it can be replaced by the top selection (*standing, on, surfboard*). Hence, it is important to consider the dependency among edges. Although causality has been considered in very recent works of other domains (Moraffah et al., 2020), they have yet to be applied in GNNs and fall short in the ability to explain GNNs. See Section 2 for an exhaustive list and full discussion.

To the best of our knowledge, causal interpretability is yet unexplored in GNNs. Here we propose a novel method, *Causal Screening*, to take a cause-effect look at feature attribution. It takes a graph of interest, along with the prediction made by a trained GNN model, and aims to return an explanatory subgraph that plausibly has the largest causal attribution on the prediction. Towards this end, Causal Screening starts from an empty set as the explanatory subgraph, and adopts the screening strategy to incrementally select edges into the subgraph, one edge at a time step. At its core is to quantify the causal attribution of an edge candidate as its individual causal effect (ICE) (Pearl, 2009), answering "What would happen to the prediction, if we add this edge into the GNN's input?". It coincides with maximizing the information-theoretic measure at each step — conditional mutual information (Ay & Polani, 2008; Janzing et al., 2013; O'Shaughnessy et al., 2020) between the edge and the prediction, conditioned on the previously selected edges. Specifically, at each step, ICE is formulated as the difference between two outcomes, where the edge had received treatment (*i.e.,* combining it with the previous selection as the GNN input) or control (*i.e.,* feeding the previous selection alone into the GNN). Last but not least, we propose an efficient version which considers the cause-effect of edge groups, rather than single edges, in order to speedup the exhaustive search. We apply Causal Screening to multiple graph classification datasets, generating qualitative results showcasing the effectiveness of our explanation subgraphs, which are more consistent, concise, and faithful to the predictions compared with existing methods. Contributions of this study can be summarized as:

- We propose a model-agnostic method, Causal Screening, to provide a cause-effect perspective for explaining GNN models, *i.e.,* uncovering causal relationships between graph features and model predictions.
- We conduct extensive experiments on three datasets, showcasing the effectiveness of our method *w.r.t.* predictive accuracy, contrastivity, and sanity checks.

## 2  RELATED WORK

**Interpretability in Non-Graph Neural Networks.** We focus preliminarily on feature attribution methods that generate post-hoc explanations for neural networks, especially convolutional neural networks (CNNs). We roughly categorize current works into two groups: (1) Studies decompose the model prediction to the input features via backpropagating the gradient-like signals. Early works like Gradient (Simonyan et al., 2014) and Gradient*Input (Shrikumar et al., 2016) directly use gradients *w.r.t.* inputs as feature importance. Some follow-on studies, such as LRP (Bach

et al., 2015) and Grad-CAM (Selvaraju et al., 2017), improve gradients by using context within layers, while Integrated Gradient (IG) (Sundararajan et al., 2017) and DeepLIFT (Shrikumar et al., 2017) work on solving the gradient saturation problem; (2) Another line approximates the decision boundary of model via feature perturbations like LIME (Ribeiro et al., 2016) or feature masking like L2X (Chen et al., 2018) and VIBI (Bang et al., 2019). LIME uses a sparse linear model as a local explainer, while L2X and VIBI learn additional neural networks as explainers to generate feature masks, which maximize the mutual information (or variation) with the masked features and the predictions. However, these works mostly focus on the statistical interpretability, thus hardly uncovering the causation of model predictions.

**Interpretability in Graph Neural Networks.** Interpretability in GNNs is less explored than that in CNNs. Thus, researchers get inspiration from the methods devised for CNNs and have made some attempts. For example, SA (Baldassarre & Azizpour, 2019) treats the gradients of GNN's loss *w.r.t.* the adjacency matrix as the attributions of edges. Pope et al. (2019) employ Grad-CAM family on GNN models. GraphLIME (Huang et al., 2020) extends LIME to graph-structured data. More recently, GNNExplainer (Ying et al., 2019) follows L2X and learns structure masking to maximize the mutual information between the subgraphs and GNN predictions. XGNN (Yuan et al., 2020) uses the graph generation to explore the class-level explanations, rather than the instance-wise. Despite the success, these methods inherit the limitations of the statistical interpretability.

**Causal Interpretability.** Causality has been considered in very recent explanation works, such as structural causal models (Chattopadhyay et al., 2019; Kim & Bastani, 2019), or counterfactual reasoning (Goyal et al., 2019). For example, based on Granger causality, CXPlain (Schwab & Karlen, 2019) learns to quantify the causal effect of a single feature by leaving it out. See Moraffah et al. (2020) for a survey. However, these approaches are mostly devised for CNNs and have yet to be applied in GNNs. Moreover, they possibly fall short in the ability to explain GNNs, due to particular challenges: (1) the relational and discrete data exist in graphs, unlike continuous features in images; and (2) the graph structures are highly entangled with the representation learning of GNNs, such that different graphs likely result in various network architectures and different scales of explanations. In this work, we take a cause-effect look at the interpretability in GNNs.

## 3 METHODOLOGY

We first frame the explanation generation for GNNs from the causality standpoint. Afterward, we detail the proposed method towards the causal interpretability by sequentially quantifying the causal attributions of single edges. Lastly, we offer an efficient version based on clusters.

### 3.1 PROBLEM FORMULATION

**Background of GNNs.** Without loss of generality, let $f : \mathbb{G} \to \{1, \cdots, S\}$ denote a trained GNN classifier in the graph classification task, where $\mathcal{G} \in \mathbb{G}$ is a graph instance involving the node set $\mathcal{V}$ and the edge set $\mathcal{E}$, along with node features and optional edge features. In most GNNs, $f$ is designed as: recursively create neural messages from neighboring nodes (or edges), incorporate these messages to enrich the representation of each ego node, then aggregate all node representations as the holistic graph representation, in order to classify it into $S$ classes.

**Task Description.** We define a graph of interest as $\mathcal{G} = \{e|e \in \mathcal{E}\}$ to highlight the structure features (*i.e.,* the existence of an edge and its endpoints). Given its prediction $\hat{y} = f(\mathcal{G})$, our goal is to identify the top $K$ causal edges and construct a convincing explanatory subgraph $\mathcal{G}_K^* = \{e_1^*, \cdots, e_K^*\} \subseteq \mathcal{G}$. As a result, $\mathcal{G}_K^*$ can be interpreted as the reason causing $f$ to make the prediction $\hat{y}$ for the instance $\mathcal{G}$. In this work, we follow the previous works (Baldassarre & Azizpour, 2019; Pope et al., 2019) to generate explanations based on structure features, and leave the identification of a subset of node features in future work.

**Causal Attribution of A Subgraph.** Technically, we obtain the causal explanatory subgraph $\mathcal{G}_K^*$ as maximizing the causal attribution metric $A(\cdot)$, which is formulated as:

$$\mathcal{G}_K^* = \arg \max_{\mathcal{G}_K \subseteq \mathcal{G}} A(\mathcal{G}_K), \quad \text{s.t.} \quad |\mathcal{G}_K| = K. \tag{1}$$

Here we frame the metric $A(\mathcal{G}_K)$ as the causal effect (CE) of $\mathcal{G}_K$, which is founded on the $do(\cdot)$ calculus (Pearl, 2009). The interventions $do(\mathcal{G} = \mathcal{G}_K)$ and $do(\mathcal{G} = \emptyset)$ mean that the GNN input $\mathcal{G}$

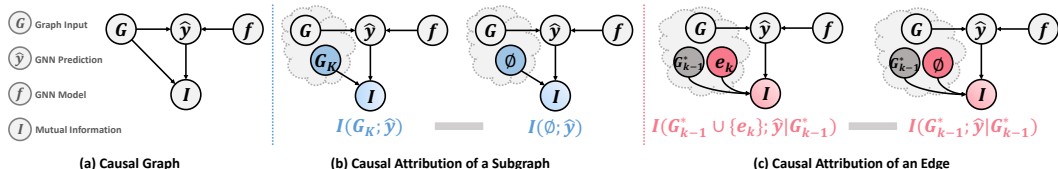

Figure 2: An illustration of our Causal Screening for obtain the causal attribution.

receives treatment (*i.e.,* feeding $\mathcal{G}_K$ into the GNN) and control (*i.e.,* feeding nothing into the GNN), respectively. The CE of $\mathcal{G}_K$ is to measure the difference between potential outcomes under treatment and control:

$$A(\mathcal{G}_K) = I(do(\mathcal{G} = \mathcal{G}_K); \hat{y}) - I(do(\mathcal{G} = \emptyset); \hat{y}), \tag{2}$$

where $I(do(\mathcal{G} = \mathcal{G}_K); \hat{y}) = I(\mathcal{G}_K; \hat{y})$ is the mutual information (Aliferis et al., 2010; Belghazi et al., 2018) between the intervened graph $\mathcal{G}_K$ and the target prediction $\hat{y}$, to calculate the change in the probability of prediction when the input graph is limited to the subgraph $\mathcal{G}_K$; analogously, $I(do(\mathcal{G} = \emptyset); \hat{y}) = I(\emptyset; \hat{y})$. As the last term $I(do(\mathcal{G} = \emptyset); \hat{y})$ is a constant, the solution of Equation 1 is equivalent to obtain a subgraph $\mathcal{G}_K^*$ with maximum predictivity to recover the target prediction $\hat{y}$. Here we summarize the attribution framework in the form of structural causal model (well-known as causal graph) (Pearl, 2009; Pearl et al., 2016; Pearl & Mackenzie, 2018), as Figure 2 (a) shows. It indicates how variables interact with each others, *e.g.,* $(\mathcal{G}, \hat{y}) \rightarrow I$. The intervention assigns the GNN input with a certain value, such that the difference caused by the treatment and control measures the causal attribution of a subgraph as shown in Figure 2 (b).

## 3.2 CAUSAL SCREENING

**Causal Attribution of An Edge.** We further move on to considering the causal attribution of an edge, due to: (1) Highlighting importance of single edges is more preferred than showing a holistic subgraph solely; (2) Directly optimizing Equation 1 is generally NP-hard (Chen et al., 2018; Zhu et al., 2020; Ying et al., 2019). Hence we propose a new method, Causal Screening, which offers an alternative solution to Equation 1 via estimating the causal attributions of edges. Specifically, it starts from an empty set as the explanatory subgraph, and incrementally selects the edges into the subgraph, one edge at a time. Thus we revisit the objective function in Equation 1 as follows:

$$e_k^* = \arg \max_{e_k \in \mathcal{O}_k} A(e_k | \mathcal{G}_{k-1}^*), \quad k = 1, 2, \cdots, K, \tag{3}$$

where $\mathcal{G}_{k-1}^* = \{e_1^*, \cdots, e_{k-1}^*\}$ is the set of the first $(k-1)$ selected edges after $(k-1)$ steps, and $\mathcal{G}_0^* = \emptyset$ at the initial step; and $e_k^*$ is the edge selected from the $\mathcal{O}_k$ at step $k$, where $\mathcal{O}_k = \mathcal{G} \backslash \mathcal{G}_{k-1}^*$ is the pool of edge candidates. Having established $\mathcal{G}_k^*$ by merging $\mathcal{G}_{k-1}^*$ and $e_k^*$, we repeat this procedure until finding all top $K$ edges as the final explanatory subgraph.

The causal attribution $A(e_k | \mathcal{G}_{k-1}^*)$ is founded upon the individual causal effect (ICE) of $e_k$, which accounts for the causal influence on the prediction and the redundancy with the previously selected edges. Given the previous selection $\mathcal{G}_{k-1}^*$, the intervention $do(\mathcal{G} = \mathcal{G}_{k-1}^* \cup \{e_k\})$ represents that the graph input receives treatment (*i.e.,* the GNN takes the combinations of $e_k$ and the previous selection $\mathcal{G}_{k-1}^*$); whereas, $do(\mathcal{G} = \mathcal{G}_{k-1}^*)$ means that the graph input receives control (*i.e.,* feeding the previously selected edges alone into the GNN). As such, the ICE is formulated as:

$$A(e_k | \mathcal{G}_{k-1}^*) = I(do(\mathcal{G} = \mathcal{G}_{k-1}^* \cup \{e_k\}); \hat{y}) - I(do(\mathcal{G} = \mathcal{G}_{k-1}^*); \hat{y}). \tag{4}$$

This coincides with conditional mutual information (Janzing et al., 2013; O'Shaughnessy et al., 2020): $A(e_k | \mathcal{G}_{k-1}) = I(\mathcal{G}_{k-1} \cup \{e_k\}; \hat{y} | \mathcal{G}_{k-1})$. Figure 2 (c) presents the ICE inference.

By calculating the ICE of each edge, we could answer the causality-related questions like "Given the previously selected edges, what is the causal effect of an edge on the target prediction?". Following the literatures of feature selection (Kohavi & John, 1997; Guyon & Elisseeff, 2003) and causality (Aliferis et al., 2010; Guyon, 2007; Athey, 2017), we categorize edges into three groups based on the causal relationships with the predictions: (1) Strongly relevant edges contain unique information pertinent to GNN predictions, which are associated with significant causal contributions. (2) Weakly relevant edges are informative but redundant, which can be inferred and replaced by the previously selected edges. Hence, these edges have varnishing (close-to-zero) causal contributions to the GNN

predictions. (3) Irrelevant edges do not bring any relevant information about predictions, whose causal contributions are zero. In a nutshell, the strongly relevant edges are more convincing and explainable than the redundant and irrelevant edges. Hence, the core of causal interpretability is to distinguish causal features (*i.e.,* strongly relevant edges) from non-causal features (*i.e.,* redundant edges and irrelevant edges). We summarize the overall algorithm in Algorithm 1 of Appendix A.1.

### 3.3 SCALING TO LARGE DATA

However, selecting edges one by one might have some limitations: (1) It hardly measures importance of feature interactions or combinations. For example, in social network analysis, group-wise patterns of users are more informative than behaviors of a single user; in genetics and genomics, block-like patterns of correlated genes work as functional groups, to determine biological properties. (2) The computational complexity of Equation 3 is $O((2|\mathcal{G}| - K)K/2)$ per graph. This will be a bottleneck of large-scale graphs involving massive edges. Hence, we provide a cluster-by-cluster screening.

**Causal Attribution of A Super-edge.** We first apply a clustering method on graph instances to map nodes into clusters. Here we resort to DiffPool (Ying et al., 2018). For a given graph $\mathcal{G}$, it outputs a node-cluster matrix $\mathbf{X} \in \mathbb{R}^{|\mathcal{V}| \times |\mathcal{C}|}$, where the $v$-th row denotes the soft cluster assignment of node $v$ to $\mathcal{C}$ clusters. By assigning every node to the cluster with the largest confidence, we are able to simplify $\mathcal{G}$ as a new super-graph — nodes in the same cluster are combined as a super-node $c \in \mathcal{C}$; as such, intra-cluster edges are grouped as a self-loop of the super-node $g = (c, c)$, while we integrate the parallel inter-cluster edges together as a super-edge $g = (c, c')$ across two super-nodes $c$ and $c'$. Here we treat these two types of super-edges equally, without distinguishing them. As a result, such super-edges not only reduce the scale of graph dramatically, but also characterize the group-wise patterns of graph (*e.g.,* user groups in social networks, functional groups in genomics).

We then employ Causal Screening on the simplified graph, to estimate the causal attributions of all super-edges $\mathcal{O} = \{g\}$ with their connected super-nodes. At each step $t \in \{1, \cdots, |\mathcal{O}|\}$, the CE of a super-edge is:

$$A(g_t | \mathcal{G}_{t-1}^*) = I(do(\mathcal{G} = \mathcal{G}_{t-1}^* \cup \{g_t\}); \hat{y}) - I(do(\mathcal{G} = \mathcal{G}_{t-1}^*); \hat{y}), \tag{5}$$

where $g_t \in \mathcal{O}_t$ is the candidate from the super-edge set $\mathcal{O}_t = \mathcal{G} \backslash \mathcal{G}_{t-1}^*$; $\mathcal{G}_{t-1}^*$ is the subgraph involving the previously selected super-edges; and the intervention $do(\mathcal{G} = \mathcal{G}_{t-1}^* \cup \{g_t\})$ is to add $g_t$ into $\mathcal{G}_{t-1}^*$ — which validates all its component edges — and feed them into the GNN model. It is worth mentioning that, as one edge belongs to one and the only one super-edge, we could eliminate its influence on the CE estimations of other super-edges. Hereafter, we can establish a CE matrix $\mathbf{H} \in \mathbb{R}^{C \times C}$ by filling the entry at the $c$-th row and $c'$-th column with the CE of the super-edge $g = (c, c')$, or leaving it as zero if no super-edge presents.

We further distribute the CEs of super-edges to the ICEs of single edges. However, it is difficult due to the challenges: (1) as the component edges of a super-edge are validated together during the intervention, there lacks the knowledge of edge sequence; (2) beyond the component edges with the connected nodes, one intervention is likely to involve some isolated nodes without any edge, thereby introducing the contribution of these isolated nodes into the CE of the super-edge. We get inspiration from the link prediction task (Koren, 2008), which exploits the function of two endpoints as a proxy to approximate the ICE of an edge. Technically, we first create the node representations via $\mathbf{Z} = \mathbf{X}\mathbf{H} \in \mathbb{R}^{|\mathcal{V}| \times |\mathcal{C}|}$, where the $v$-th row $\mathbf{z}_v$ is the representation of node $v$ that describes the contribution distributions over different clusters. Then, for an edge $e = (v, v')$, we simply apply the inner product on the representations of endpoints $v$ and $v'$ to approximate $e$'s attribution:

$$A(e) = \mathbf{z}_v^\top \mathbf{z}_{v'}. \tag{6}$$

Intuitively, if two endpoints have similar contribution distributions over clusters, the edge is highly likely to be influential to the information flow along with the graph structure and contribute more to the GNN prediction. Through this way, we are able to estimate the ICEs of single edges and retrieve the top-$K$ edges as the explanatory subgraph. This hierarchical fashion essentially reflects the modeling of GNNs: single nodes pass information to each others along with the graph structure, enhancing the group-wise patterns; thereafter, the patterns are used to classify the graph types. Moreover, this approximation allows us to get results significantly faster than exact calculations, as its time complexity is $O((2|\mathcal{O}| + 1)|\mathcal{O}|/2 + |\mathcal{C}|^2)$, where $|\mathcal{O}|$ and $|\mathcal{C}|$ are the numbers of super-edges and clusters, respectively. The algorithm is summarized in Algorithm 2 of Appendix A.1.

### 3.4 GENERALIZATION TO OTHER FEATURES

In many real-world scenarios like social networking (Tang et al., 2015) and recommendation systems (Zhang & Chen, 2018), only ID information is usually assigned with nodes or edges, without any pre-existing feature. Hence, the structure features (*i.e.,* presence of edges, connections between nodes) attract more attention than other features (*e.g.,* node features). We follow the previous studies (Yuan et al., 2020; Pope et al., 2019; Baldassarre & Azizpour, 2019) and build explanations upon the edge importance, which are applicable to general graph data. Furthermore, as a general idea, Causal Screening can be easily generalized to other feature importance (*e.g.,* presence of nodes, features of nodes) by simply changing ICEs of edges to ICEs of target features.

## 4 EXPERIMENTS

We aim to investigate the research question: How do the explanations of Causal Screening perform?

### 4.1 EXPERIMENTAL SETTINGS

We conduct comparative experiments on several various benchmark datasets over different GNN architectures, performing both quantitative and qualitative analyses to assess explanations.

**Datasets with GNNs.** We consider the graph classification task on three real-world domains:

- **Molecule graph classification.** We use the Mutagenicity dataset (Kazius et al., 2005; Riesen & Bunke, 2008), where $4,337$ molecule graphs are categorized into two different classes based on their mutagenic effect on a bacterium. The GNN model used is the graph isomorphism network (GIN) (Xu et al., 2019; Hu et al., 2020).
- **Social network classification.** REDDIT-MULTI-5K (Yanardag & Vishwanathan, 2015) is used, where $4,999$ graphs are labeled into five different classes based on the topics of question/answer communities. A higher-order graph neural network (k-GNN) (Morris et al., 2019) is trained as the target classifier model.
- **Scene graph classification.** We follow the previous work (Pope et al., 2019) and extract parts of the data from Visual Genome (Krishna et al., 2017), where each image is paired with a scene graph. In each scene graph, nodes are objects associated with a region of the image, while edges describe the relationships between two objects. Here we select $4,443$ (images, scene graphs) pairs which are labeled with five classes: stadium, street, farm, surfing, forest. We then train a APPNP model (Klicpera et al., 2019) on this dataset to classify the scene graphs into these classes.

Following prior studies (Pope et al., 2019; Ying et al., 2019), we partition each dataset into the training, validation, and testing parts with the ratio of $80\%:10\%:10\%$. All graph instances are directed. The GNN models are trained on the training dataset, while we run the explanation methods five times and report the average classification and explanation performance on the testing dataset. See Table 4 in Appendix A.2.1 for the statistics of datasets and the configurations of GNN models.

**Alternative Baseline Approaches.** We consider the state-of-the-art explanation methods for GNNs: SA (Baldassarre & Azizpour, 2019), Grad-CAM (Pope et al., 2019), and GNNExplainer (Ying et al., 2019), as our baselines. Moreover, we transfer and extend IG (Sundararajan et al., 2017), DeepLIFT (Shrikumar et al., 2017), and CXPlain (Schwab & Karlen, 2019) to the graph domains.

**Evaluation Metrics.** It is of crucial importance to evaluate the effectiveness of explanation methods. Although the ground truth of explanations is usually unavailable, previous works have proposed some metrics to quantitatively assess the explanations. Here we consider the following metrics (see Appendix A.2.3 for the detailed formulations):

- **Accuracy** (Chen et al., 2018; Liang et al., 2020) (ACC) reflects the consistency between the predictions based on the whole graph and the explanatory subgraphs.
- **Contrastivity** (Pope et al., 2019) (CST) is to capture the intuition that class-discriminative explanations should differ among classes.
- **Sanity Check** (Adebayo et al., 2018) (SC) is to test whether the explanations are dependent on the target model. The attributions on the trained GNN model are compared with that on

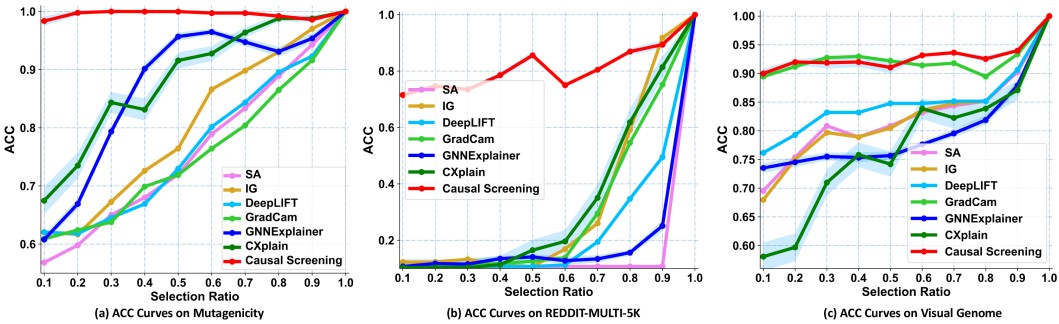

Figure 3: Accuracy curves of explanation methods over selection ratios. Best viewed in color.

Table 1: Overall Performance Comparison for explanation methods *w.r.t.* accuracy and contrastivity metrics. The best performance is highlighted with *, while the strongest baselines are underlined.

| | Mutagenicity | | | REDDIT-MULTI-5K | | | Visual Genome | | |
|---|---|---|---|---|---|---|---|---|---|
| | ACC-AUC↑ | CST↓ | SC↓ | ACC-AUC↑ | CST↓ | SC↓ | ACC-AUC↑ | CST↓ | SC↓ |
| SA | 0.767 | 0.975 | 0.221 | 0.196 | 0.583 | 0.278 | 0.829 | 0.462 | 0.351 |
| IG | 0.807 | 0.990 | 0.365 | 0.354 | 0.894 | 0.268 | 0.826 | 0.438 | 0.345 |
| DeepLIFT | 0.775 | 0.993 | 0.258 | 0.268 | 0.667 | 0.411 | 0.852 | 0.434 | 0.370 |
| Grad-CAM | 0.764 | 0.768 | 0.254 | 0.330 | 0.528 | 0.327 | 0.917 | 0.426 | 0.755 |
| GNNExplainer | 0.872 | 0.690 | 0.124* | 0.228 | 0.911 | 0.198 | 0.802 | 0.417 | 0.267* |
| CXPlain | 0.886 | 0.587 | 0.327 | 0.356 | 0.664 | 0.722 | 0.775 | 0.320 | 0.361 |
| Causal Screening | 0.995* | 0.155* | 0.278 | 0.825* | 0.168* | 0.167* | 0.930* | 0.315* | 0.316 |

an untrained GNN with randomly initialized parameters. Similar attributions infer that the explanation method is insensitive to properties of the model, thus failing to pass the check.

By default, we employ the standard Causal Screening on Mutagenicity and Visual Genome, while the cluster-based version on REDDIT-MULTI-5K (with $|\mathcal{C}| = 10$) due to its larger scale of graphs.

## 4.2 EVALUATION OF EXPLANATIONS

**Quantitative Analyses.** We begin with the comparison *w.r.t.* accuracy, where the ACC curve over different selection ratios is presented in Figure 3 and the area under curve is reported as ACC-AUC in Table 1. We find that: (1) Causal Screening achieves significant improvements over the compared approaches. For example, when only 10% edges are selected in Mutagenicity and REDDIT-MULTI-5K datasets, its ACC scores outperform others' by a large margin. (2) The edges selected by our method are indeed important and faithful to the GNN models. Especially, in the Mutagenicity and Visual Genome datasets, our method shows nearly optimal fidelity (ACC-AUC scores are 0.995 and 0.930, respectively), which is validated by the GNN models. (3) Having selected the strongly relevant edges, adding more noncausal or redundant edges has only a negligible impact on the classifier accuracy. This justifies the rationality and effectiveness of our conditional causal attributions. (4) It verifies the superiority of causal interpretability over to the statistical interpretability derived from other approaches (say, SA family and GNNExplainer). These methods may attribute importance to the confounding associations, rather than uncovering the causation. Hence, their explanations are unsatisfactory and less accurate. (5) Our method performs better than CXPlain, suggesting that screening to combine an edge with the previous selection estimates the causal effects of edges more accurately, compared to leaving a single edge out. This emphasizes the rationality of considering edge dependency and confounding factors. In a nutshell, taking advantage of causal attributions, Causal Screening is able to generate faithful explanations to GNN models.

We move on to the contrastivity reported in Table 1, to check whether explanations differ when the predictions of interest are different. We find that Causal Screening achieves the lowest contrastivity scores consistently. For a graph, the rank correlation for our explanations in different classes is low — that is, the explanations are diverging from each others, such that a clear boundary is easy to identify. For example, in the Mutagenicity dataset, different chemical groups are selected to explain the mutagenic and non-mutagenic labels. It evidently shows that our method is able to offer class-discriminative explanations.

We then focus on the sanity checks, where the results are summarized in Table 1. We observe that: (1) Causal Screening achieves low SC scores and passes the checks, indicating that the attributions

Table 2: A comparison of compute time (in seconds) per explanation.

|  | SA | IG | DeepLIFT | Grad-CAM | GNNExplainer | CXPlain | Causal Screening |
|---|---|---|---|---|---|---|---|
| Mutagenicity | 0.011 | 0.221 | 0.060 | 0.010 | 2.57 | 1.74 | 1.83 |
| REDDIT-MULTI-5K | 0.008 | 0.172 | 0.047 | 0.008 | 2.14 | 13.4 | 2.52 |
| Visual Genome | 0.015 | 0.282 | 0.062 | 0.015 | 2.71 | 1.77 | 0.326 |

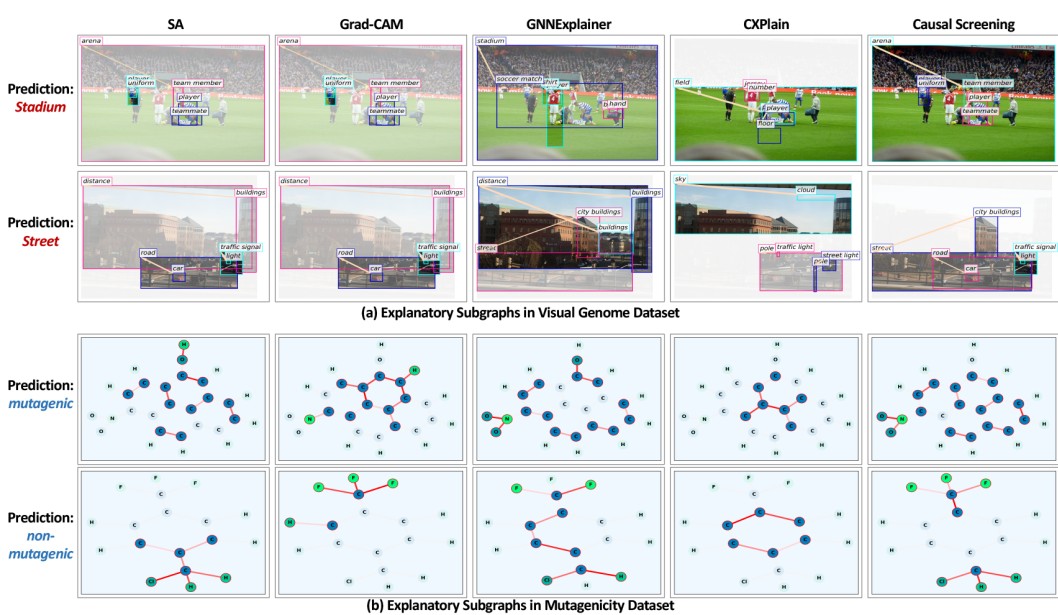

Figure 4: Selected explanations for each explanation methods, where the top 20% edges are highlighted. Note that some edges have the same nodes. In Visual Genome, the objects involved in the edges are blurred based on the edge attributions; meanwhile, in Mutagenicity, a darker color of a bond indicates the larger attribution for the prediction. Best viewed in color.

on the trained and untrained GNNs differ significantly. This sensitivity reveals that our explanations are dependent on the target model. (2) Grad-CAM achieves higher SC scores than SA substantially. This is consistent with the findings in (Adebayo et al., 2018). Insensitivity to the model parameters makes these methods fail to explain the inner working of GNNs reliably. This evaluation again supports the rationality of our method.

**Qualitative Analyses.** We present visual inspections in Figure 4 and observe that: (1) Our method not only selects more influential edges that possibly cause the prediction, but also ensures the briefness of edge combinations. For example, when explaining the prediction of *Stadium*, the most convincing edge (*team member, in, arena*) is only ranked at the top of explanations by Causal Screening; moreover, our method treats (*player, in, uniform*) as the second edge, rather than the confounding associations like (*shirt, on, player*) selected by GNNExplainer or the redundant information like (*player, lying down on, floor*) and (*player, in field*) simultaneously captured by CXPlain. When explaining the prediction of *Street*, Causal Screening selects (*city buildings, on, street*), while SA and Grad-CAM fail. This again verifies the superiority of the causal interpretability to the statistical interpretability. (2) In Mutagenicity, Causal Screening is able to identify functional groups (which are not precisely defined but evaluated by the domain experts). For the 1-Nitro-5-Pyrene molecule labeled as mutagenic, our method highlights the adjacent carbon rings and the $NO_2$ chemical group, which are both responsible for the mutagenic property (Debnath et al., 1991). Whereas, the baselines fail since they are possibly misled by some noncausal chemical bonds. As for the prediction of non-mutagenic, our method assigns large causal attributions to chlorine (Cl) and fluorine (F), which is consistent to the findings that the combination of Cl, bromine (Br), and F usually leads to non-mutagenic predictions (Yuan et al., 2020). In contrast, Grad-CAM ignores Cl and is distracted by the carbon-hydrogen (C-H) bond, and CXPlain gives high important scores to irrelevant bonds but excludes C-Cl and C-F bonds out.

**Time Complexity Analyses.** We also report the actual runtime of explanation methods in Table 2 on a GPU of NIVIDA Quadro T2000 4G. In Mutagenicity and Visual Genome where the graphs are

Table 3: Influence of clusters for cluster-based Causal Screening in the REDDIT-MULTI-5K dataset.

| Cluster# | - | 5 | 10 | 15 | 20 | 25 | 30 |
|---|---|---|---|---|---|---|---|
| ACC-AUC↑ | 0.508 | 0.816 | 0.825 | 0.823 | 0.811 | 0.819 | 0.815 |
| CST↓ | 0.078 | 0.086 | 0.168 | 0.187 | 0.098 | 0.062 | 0.066 |
| SC↓ | 0.082 | 0.162 | 0.167 | 0.162 | 0.187 | 0.175 | 0.182 |
| Compute Time | 72.38 | 1.010 | 2.520 | 6.501 | 8.810 | 12.33 | 15.80 |

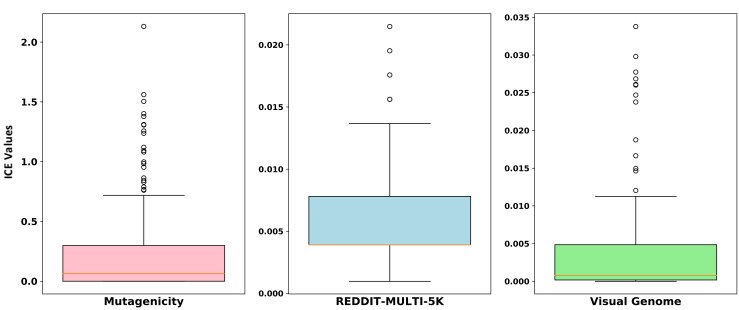

Figure 5: Distribution of ICE values.

of a small scale, the standard Causal Screening generates explanations faster than GNNExplainer; meanwhile, in REDDIT-MULTI-5K where relatively large-scale edges exist in graphs, our cluster-based version has a comparable time cost to GNNExplainer and CXPlain.

### 4.3 STUDY OF CAUSAL SCREENING

**Influence of Cluster Numbers.** Table 3 presents the performance changes of the cluster-based Causal Screening over the cluster numbers. (1) The computational cost drops significantly when the cluster number changes from 30 to 5, but all of them are much efficient than the standard version (*i.e.,* the first column). (2) Surprisingly, the cluster-based version outperforms the standard version by a large margin *w.r.t.* ACC-AUC. We attribute such improvements to the clustering, which not only discovers the group-wise patterns of nodes, but also introduces the interactions among single edges into the causal attributions. In essence, the cluster-based approximation works as a structural function between the edge and mutual information variables (Pearl et al., 2016) and yields state-of-the-art results. This verifies the rationality and effectiveness of our cluster-based approximation.

**Showcases of Causal Attributions.** For the top 20% edges selected by Causal Screening, we produce statistics of their ICE values and display their distributions in Figure 5. In particular, each boxplot shows five measures of central tendency: minimum, first quartile, median, third quartile, and maximum. We find that, the ICE values of causal edges (*i.e.,* strongly relevant) are much larger than that of non-causal edges (*i.e.,* redundant or irrelevant). For example, in Mutagenicity, the causal edge has an ICE value within the range of 1 to 3, while the ICE of the non-causal approaches zero.

In a nutshell, through quantitative evaluations and visual inspections, we show that Causal Screening is able to generate high-quality post-hoc explanations reflecting the causation of GNN predictions. Hence, the explanations as the distilled knowledge can in turn help us inspect the GNN models and improve them. For instance, by emphasizing the informative structures, we can prevent GNNs from being distracted by redundant or useless information, further improve the model designs.

## 5 CONCLUSION

In this paper, we have explored the causal interpretability of graph neural networks, and proposed a model-agnostic method, Causal Screening, to identify the most influential edges and generate post-hoc explanations for model predictions. We make in-depth analyses on causal explanations *w.r.t.* quantitatively evaluations and qualitative inspections, and observe that: causal attributions of edges are more reliable than gradient-like signals or feature masks obtained from the statistical perspective. This work represents an initial attempt to exploit causality in GNN's explanations. In the future, we would like to consider more causality concepts, such as counterfactual reasoning to analyze the key features which make predictions different. Moreover, we will extend our work to class-wise explanations like XGNN (Yuan et al., 2020).

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

# A APPENDIX

## A.1 ALGORITHMS

Here we present the algorithms for Causal Screening and the cluster-based version in Algorithms 1 and 2, respectively.

---

**Algorithm 1:** Causal Screening

---

**Input:** $\mathcal{G}$: graph instance, $f$: target GNN model, $\hat{y}$: predicted label of $\mathcal{G}$, $H$: conditional entropy, $K$: size of explanations;

**Output:** $\mathcal{G}_K^*$: explanatory subgraph;

$\mathcal{G}_0^* \leftarrow \emptyset$;

**foreach** $k \in \{1, 2, \cdots, K\}$ **do**

    $maxA \leftarrow 0$;

    $\mathcal{O}_k \leftarrow \mathcal{G} \backslash \mathcal{G}_{k-1}^*$;

    **foreach** $e_k \in \mathcal{O}_k$ **do**

        $I(\mathcal{G}_{k-1}^*; \hat{y}) \leftarrow H(\hat{y}|\mathcal{G}_{k-1}^*)$;

        $I(\mathcal{G}_{k-1}^* \cup \{e_k\}; \hat{y}) \leftarrow H(\hat{y}|\mathcal{G}_{k-1}^* \cup \{e_k\})$;

        $A(e_k|\mathcal{G}_{k-1}^*) \leftarrow I(\mathcal{G}_{k-1}^* \cup \{e_k\}; \hat{y}) - I(\mathcal{G}_{k-1}^*; \hat{y})$ ;

        **if** $A(e_k|\mathcal{G}_{k-1}^*) > maxA$ **then**

            $e_k^* \leftarrow e_k$;

            $maxA \leftarrow A(e_k|\mathcal{G}_{k-1}^*)$;

        **end**

    **end**

    $\mathcal{G}_k^* \leftarrow \mathcal{G}_{k-1}^* \cup e_k^*$;

**end**

---

**Algorithm 2:** Cluster-based Causal Screening

---

**Input:** $\mathcal{G}$: graph instance, $f$: target GNN model, $\hat{y}$: predicted label of $\mathcal{G}$, $H$: conditional entropy, $K$: size of explanations, $p$: clustering method, $C$: number of clusters;

**Output:** $\mathcal{G}_K^*$: explanatory subgraph

$\mathbf{X} \leftarrow p(\mathcal{G}, C)$ is the node-cluster assignment matrix;

$\mathcal{O} = \{g\}$ is the set of super-edges built upon the clusters;

$\mathbf{H} \in \mathbb{R}^{C \times C}$ is the attribution matrix, after performing Algorithm 1 over the super-edges;

$\mathbf{Z} = \mathbf{XH}$ is the attribution embedding matrix;

$\mathcal{G}_K^*$ is the top-$K$ edges with the highest attribution scores based on Equation 6;

---

## A.2 EXPERIMENTS

### A.2.1 DATASETS WITH GNNS.

We summarize the statistics of the datasets in Table 4. In particular, we use the original Mutagenicity and REDDIT-MULTI-5K datasets[1]. As for the Visual Genome dataset, we follow the previous work (Pope et al., 2019) to construct the subset from the whole dataset[2] (Krishna et al., 2017). Different from Pope et al. (2019) that focuses on two binary classification problems, we define a multi-class classification problem: stadium, street, farm, surfing, forest. By querying the whole dataset via a set of keywords, we combine the images returned for a class and discard the images across classes.

We build GNN models for these datasets via the PyTorch Geometric library[3] (Fey & Lenssen, 2019). For Mutagenicity, we treat the given labels of nodes and edges as their pre-existing features, apply 1-layer MLPs to project them into a common space, and then build a 2-layer GIN to perform the graph learning, which follows a average pooling and a 2-layer MLP to do classification. For REDDIT-

---

[1] https://ls11-www.cs.tu-dortmund.de/staff/morris/graphkerneldatasets.
[2] https://visualgenome.org/.
[3] https://pytorch-geometric.readthedocs.io/en/latest/.

Table 4: Statistics of three datasets and GNN models.

| Dataset | Graphs# | Classes# | Avg. Nodes# | Avg. Edges# | GNN Models | Layers# | Accuracy |
|---|---|---|---|---|---|---|---|
| Mutagenicity | 4,337 | 2 | 30.32 | 30.77 | GIN | 2 | 0.806 |
| REDDIT-MULTI-5K | 4,999 | 5 | 508.52 | 594.87 | k-GNN | 3 | 0.978 |
| Visual Genome | 4,443 | 5 | 35.32 | 18.04 | APPNP | 2 | 0.640 |

MULTI-5K, a 3-layer k-GNN is built upon the degree features of nodes, and a average pooling is adopted to obtain the graph representation, which follows a 2-layer MLP to classify. For Visual Genome, we first apply CNNs on the regional images to generate features for the object nodes, and then use a 2-layer APPNP followed by a 2-layer MLP as the classifier model. We carefully train these models, and report their best performance in Table 4.

### A.2.2 IMPLEMENTAL DETAILS

We implement Causal Screening in PyTorch, and will release all datasets and codes upon acceptance. All the experiments are conducted on the GPU of NIVIDA Quadro T2000 4G. Without specification, we employ the standard Causal Screening (*i.e.,* Algorithm 1) on Mutagenicity and Visual Genome, while the cluster-based version (*i.e.,* Algorithm 2) on REDDIT-MULTI-5K (with $|\mathcal{C}| = 10$) due to its larger scale of graphs. The empirical results are reported in Table 1. Moreover, we report the comparison between the standard and cluster-based versions, as well as the influence of cluster numbers in Table 3.

For the baseline SA, we follow the codes released by Baldassarre & Azizpour (2019). In terms of IG, we create a new graph, which remains all node features but remove all edge features as the graph reference; moreover, the path steps of IG is 20. As for DeepLIFT, we further remove node features as the reference in order to solve the saturation problem in graph-structured data. For CXPlain and GNNExplainer that involve additional explainer networks, we apply a grid search to tune their hyper-parameters, and get the optimal settings: the learning rate is set as $0.01$ in CXPlain and GNNExplainer, the regularization term and the weight of mutual information of GNNExplainer are set as $0.05$ and $0.5$, respectively.

### A.2.3 EVALUATION METRICS.

We adopt three widely-used metrics to quantitatively evaluate whether the explanations faithfully reflect the relationship between the input and model output: accuracy, contrastivity, and sanity check. Here we do not consider the deletion or removal of edges as the evaluation metric, as suggested by Hooker et al. (2019).

**Accuracy** (Chen et al., 2018; Liang et al., 2020) (ACC). This reflects the consistency between the predictions based on the whole graph and the explanatory subgraphs. More formally, given a graph, we evaluate whether the GNN model truly uses the explanatory subgraph derived from an explanation method to generate the prediction:

$$\text{ACC}(\mu) = \mathbb{E}_{\mathcal{G} \sim \mathbb{G}}[\mathbb{I}(f(\mathcal{G}), f(\mathcal{G}_K^*))], \qquad (7)$$

where $\mu$ is the selection ratio (*e.g.,* $5\%$), $K = \mu \times |\mathcal{G}|$ is the size of explanatory subgraph; $\mathbb{I}(\cdot)$ is the indicator function to check whether $f(\mathcal{G})$ equals to $f(\mathcal{G}_K^*)$.

**Contrastivity** (Pope et al., 2019) (CST). This is to capture the intuition that a reasonable explanation method should entail differing explanations with different predicted labels. Here we formulate it as:

$$\text{CST} = \mathbb{E}_{\mathcal{G} \sim \mathbb{G}} \mathbb{E}_{s \neq \hat{y}}[|\rho(\Phi(\mathcal{G}, s), \Phi(\mathcal{G}, \hat{y}))|], \qquad (8)$$

where $s$ means we permute the label $\hat{y}$ of $\mathcal{G}$ being interpreted; $\Phi(\mathcal{G}, \hat{y})$ is the attribution scores of all edges; $|\rho(\cdot)|$ is the Spearman rank correlation with the absolute value to measure the invariance between the edge attributions, in response to the label changes.

**Sanity Check** (Adebayo et al., 2018) (SC). We use the model randomization test to evaluate whether the explanations are dependent on the target model. Specially, this test compares the attributions on the trained GNN model $f$ with the attributions on an untrained GNN model $\tilde{f}$ with randomly initialized parameters. Here we frame it as the rank correlation between these two attributions:

$$\text{SC} = \mathbb{E}_{\mathcal{G} \sim \mathbb{G}}[|\rho(\Phi(\mathcal{G}, f(\mathcal{G})), \Phi(\mathcal{G}, \tilde{f}(\mathcal{G})))|]. \qquad (9)$$

Similar attributions infer the explanation method is insensitive to properties of the model, thus failing to pass the check.

