# OpenReview forum: "Causal Screening to Interpret Graph Neural Networks"
_ICLR.cc/2021/Conference — Reject_

### Official Review · AnonReviewer2 · 2020-10-27
**Official Blind Review #2**

**Rating:** 5
**Confidence:** 4

**Review:**

This work proposes to explain graph neural networks from a causal effect view. The proposed method, Causal Screening, iteratively adds edges into the explanatory subgraph. The goal is to maximize the $do(\cdot)$ calculus, which tries to select an edge such that the prediction probability when feeding into GNNs is close to the original prediction. Experimental results show that the proposed method can outperform other comparing methods.

Strengths:
+ Explaining GNNs is very important and interesting. This work focuses on the causal attribution view, which is a good direction.
+ The proposed “Causal Attribution of A Super-edge” is very useful for large-scale graphs, which can reduce the computational cost.
+ Experimental results show that the proposed method can outperform other methods for some metrics.

Weaknesses:
- The first concern is the motivation of this work. The motivation is coming from the running example in the introduction. It claims that other method leads to confounding association that (shorts, on, man), and (man, has, hand) are correlated with the prediction, rather than causing it. However, this is from the human’s view, not the model’s view. For interpretations, the method needs to explain the model. How do we know (shorts, on, man), and (man, has, hand) are not causing the prediction?
- The proposed method is very straightforward, which is a simple application of the individual causal effect. The technical contribution may not be enough for ICLR publication.
- The proposed method now only considers the edge importance while other comparing methods consider the importance of edges, nodes, and features.
- The computational cost of the proposed method is very high. For each step, it needs to call the GNN multiple times to select a good edge from the candidate.
- The GNN-explainer can obtain much better sanity check scores with the model randomization test, which means the proposed method is less dependent (less faithful) to the model compared with GNN-explainer.

I am willing to adjust my score if my concerns are properly addressed.

=====Update after rebuttal=====

I have read the authors' rebuttal. I am increasing my score to 5 as some of my concerns are addressed.

1. I do not agree with the responses for the "Motivation --- Causality in Running Example". First, the GNNExplainer tries to maximize to mutual information, which means important edges for the prediction should be kept. Then why does the GNNExplaienr suffer from the confounding association? Second, "individually feeding the top edges into the target GNN" is not convincing. "An  edge is important for the prediction“ does not mean “the prediction score is high when feeding individually”.

2. I still believe the proposed method is very straightforward and the novelty is limited.

---

> ### Author Response · Authors · 2020-11-19
> **Response to reviewer #2 (part 1 of 2)**
>
> We would like to thank the reviewer's valuable feedback. Here are our responses to your comments.
>
> [Q1: Motivation --- Causality in Running Example]
> Thanks for your insightful feedback. We claim from several dimensions:
> 1. Prior studies [1,2,3] have shown that gradient- and attention-based explanations primarily capture correlations rather than causation, thus do not capture the causal influence of an input on a particular output. This is due to the fact that these explainable models cannot estimate how intervening a feature would change the predictions. Hence, as SA and GNNExplainer rely on gradients and attentions to highlight (shorts, on, man) and (man, has, had) respectively, these edges theoretically reflect the correlations with the prediction, instead of the causation.
> 2. Empirically, when individually feeding the top edges into the target GNN (e.g., (shorts, on, man), (man, has, had), (standing, on surfboard) in the running example), the predictive accuracies are $0.671$, $0.679$, and $0.852$, respectively. Clearly, (shorts, on, man) and (man, has, had) are less contributions (or less faithful) to the prediction, compared with (standing, on surfboard). Moreover, when combing these two edges with (standing, on surfboard), the change in prediction accuracy is slight. This suggests that, these two edges are weakly relevant to the prediction, conditioned on the strongest cause (standing, on surfboard).
> 3. Furthermore, we would like to emphasize that the running example is derived from the model view --- for the same GNN being explained, three explainable models start from associational and causal standpoints, and arrive at different explanations. We found that causal explanations are more understandable and consistent with human cognition.
>
> In a nutshell, from the theoretical and empirical perspectives, we can safely verify the rationality of the running example and our motivation.
>
> [Q2: Simple Estimation of Individual Causal Effect]
> Thanks for your comments. Here we would like to highlight the novelty of our work:
> 1. To the best of our knowledge, this work is the first to account causal interpretability in GNNs.
> 2. Technically, we estimate the individual causal effect of an edge as the conditional mutual information, rather than the changes of prediction values. The table below displays the performance comparison. Moreover, we proposed a cluster-based variant, which can be efficiently applied on large-scale graphs.
> 3. While our model might by straightforward in retrospect, it achieves significant improvements over the state-of-the-art models w.r.t. explanation accuracy and contrastivity. Furthermore, our model is easy to implement and reproduce, and broadly applicable to various real-world scenarios.
>
> [Q3: Importance of Edges, Nodes, \& Features]
> Thanks for your feedback. In many real-world scenarios like social networking [4] and recommendation systems [5], only ID information is usually assigned with nodes or edges, without any pre-existing feature. Hence, the structure features (i.e., presence of edges, connections between nodes) attract more attention than other features (e.g., node features). We follow the previous studies [6,7,8] in GNN explanations to build explanations upon the structure features merely (here is the presence of an edge), which are applicable to general graph data. Furthermore, as a general idea, Causal Screening can be easily generalized to other feature importance (e.g., presence of nodes, features of nodes), by simply changing ICEs of edges to ICEs of target features.
>
> [Q4: Time Complexity]
> Thanks for the comments.
> 1. In our implementation, we parallelize the calls of GNN inference --- that is, integrate the calls in every iteration into a mini-batch --- such that only $K$ calls are required to select $\mathcal{G}_{K}$. It hence greatly reduces the compute time.
> 2. We proposed a cluster-based Causal Screening to dramatically reduce the time complexity, and reported the time cost in Table 2. As we can see, in REDDIT-MULTI-5K, our method has a comparable complexity to GNNExplainer and CXPlain, while being more efficient than GNNExplainer and CXPlain in Mutagenicity and Visual Genome datasets.
>
> In a nutshell, the time complexity is not a bottleneck of our model.
>
> Reference
> 1. Causal explanations for model interpretation under uncertainty. NeurIPS'2019
> 2. Causal interpretability for machine learning-problems, methods and evaluation. SIGKDD Explorations'2020
> 3. Neural network attributions: A causal perspective. ICML'2019
> 4. LINE: large-scale information network embedding. WWW'2015
> 5. Link prediction based on graph neural networks. NeurIPS'2018
> 6. XGNN: towards model-level explanations of graph neural networks. KDD'2020
> 7. Explainability methods for graph convolutional neural networks. CVPR'2019
> 8. Explainability techniques for graph convolutional networks. 2019.

---

> > ### Author Response · Authors · 2020-11-19
> > **Response to reviewer #2 (part 2 of 2) due to the space limitation**
> >
> > [Q5: Sanity Checks]
> > Thanks for your valuable comments. We would like to clarify that:
> > 1. As [1] suggests, Sanity Check only reflects the dependence of the explanation and target model, rather than the faithfulness of explanation. If one explanation method fails the test (i.e., the Sanity Check scores are invariant w.r.t. model randomization), we can safely reject the method in practice.
> > 2. As prior studies [2,3] show, Explanation Accuracy is able to measure the faithfulness of explanations. Clearly, in Table 1 of the revision, our Causal Screening outperforms GNNExplainer w.r.t. Explanation Accuracy, indicating that our method is more faithful than GNNExplainer.
> > 3. Moreover, we conducted an additional experiment by considering a naive baseline, Random, which randomly selects edges. The table below shows the performance comparison. Clearly, Random achieves the lowest scores w.r.t. Sanity Check, but performs poor w.r.t. Explanation Accuracy. Hence, based on the results, we can conclude that Causal Screening and GNNExplainer safely pass the Sanity Check.
> >
> > |  	| Mutag. (s) 	| Mutag. (s) 	| Mutag. (s) 	| REDDIT. (c) 	| REDDIT. (c) 	| REDDIT. (c) 	| Visual Genome (s) 	| Visual Genome (s) 	| Visual Genome (s) 	|
> > |-	|-	|-	|-	|-	|-	|-	|-	|-	|-	|
> > |  	| ACC-AUC 	| CST 	| SC 	| ACC-AUC 	| CST 	| SC 	| ACC-AUC 	| CST 	| SC 	|
> > | Random 	| 0.701 	| 0.026 	| 0.123 	| 0.272 	| 0.020 	| 0.021 	| 0.764 	| 0.343 	| 0.220 	|
> > | GNNExplainer 	| 0.872 	| 0.690 	| 0.124 	| 0.228 	| 0.911 	| 0.198 	| 0.802 	| 0.417 	| 0.267 	|
> > | Causal Screening 	| 0.995 	| 0.155 	| 0.278 	| 0.823 	| 0.187 	| 0.162 	| 0.930 	| 0.315 	| 0.316 	|
> >
> > Reference
> > 1. Sanity checks for saliency maps. NeurIPS'2018
> > 2. Learning to explain: An information-theoretic perspective on model interpretation. ICML'2018
> > 3. Adversarial inﬁdelity learning for model interpretation. KDD'2020

---

### Official Review · AnonReviewer4 · 2020-10-27
**The paper proposes a method to explain predictions made by GNN. The explanations are presented as a graph. The paper is well-written and reports interesting results.**

**Rating:** 7
**Confidence:** 3

**Review:**

Summary: The paper introduces a novel method called Causal screening which takes a graph and the prediction made by a GNN, and returns an explanatory subgraph. The method aims to explain GNN models. To be precise, it starts from an empty set as the explanatory subgraph, and incrementally adds the edges, testing them for the individual causal effect.

Strengths: The paper is well-written. It addresses the limitation of many interpretable methods which are based on statistical interpretability, and ignore causal relations.
I also like the idea to do cluster-by-cluster screening where edges across two clusters serve as a super-edge. I also agree that causal attributions of edges of edges are more reliable than information from gradients, often used to explain models.

I also liked the running example which clarifies the contributions.

Weaknesses: It is interesting to see what is the value of eq. 4 in practical applications.  I can imagine that the value can be quite small, and probably some threshold is needed. I see that doing clustering can leverage this effect. However, doing clustering and working on the clusters can provide quite different results that working on separate edges. One edge which is quite strong can impact the behaviour of the whole cluster.
I guess that it is interesting to study individual impact of edges on the clusters.

---

> ### Author Response · Authors · 2020-11-19
> **Response to reviewer #4**
>
> Many thanks for your feedback. We are glad you are pleased with this work. Here are our responses to your comments.
>
> [Q1: Showcase of Causal Attributions]
> Thanks for the good suggestion. In the revised manuscript, we produced statistics of the edges' ICE values, and displayed their distribution in boxplots of Figure 5. In particular, each boxplot shows five measures of central tendency: minimum, first quartile, median, third quartile, and maximum. We found that the ICE values of causal edges (i.e., strongly relevant) are much larger than that of non-causal edges (i.e., redundant or irrelevant). For example, in the Mutagenicity dataset, the causal edge has an ICE value within the range of 1 to 3, while the ICE of non-causal edge approaches zero.
>
> [Q2: Impact of Individual Edges on Clusters]
> Thanks for your valuable feedback. The causal attribution of a super-edge is composed of the causal attributions of single edges.
> Hence, we distribute the attribution of a super-edge to representations of nodes, which serve as a proxy to approximate the ICE of an edge. We will explore the influence of individual edges on super-edges in future work.

---

### Official Review · AnonReviewer1 · 2020-10-29
**Clear presentation; interesting idea; but the formula does not seem correct**

**Rating:** 5
**Confidence:** 4

**Review:**

This paper proposes a method, called Causal Screening, to improve the interpretability of GNN. Basically, the proposed method adds each edge in GNN in a greedy way by evaluating its causal effect on the prediction. The experimental results show the improved performance of the proposed method compared to others.

Pros:
Overall, the presentation is clear and the idea is interesting.

Cons:
However, I have the following two concerns.
1. First, Eq. (1) and Eq. (3) do not guarantee to achieve the same graph, because Eq. (3) adds each edge in a greedy way, which may result in a suboptimal graph. To mitigate this issue, besides the forward phrase, the authors may consider adding a backward phase to remove spurious edges.
2. Second, the formula for causal effect estimation (Eq. (2) and Eq. (4)) seems strange to me. It is different from the normal formula of the potential-outcome estimation. I do not understand why the authors estimate the mutual information between a constant (a particular graph) and the prediction.

If the authors can solve the above problems, I will increase the score.

Furthermore, to better show the efficacy of the proposed method, in the experiments, the authors should vary the graph size and graph density.


Post-rebuttal:
Thanks for the feedback. By optimizing Eq. (3), the learned graph is not guaranteed causal, so I would like to suggest the authors not emphasizing "causal".

---

> ### Author Response · Authors · 2020-11-19
> **Response to reviewer #1**
>
> Many thanks for your feedback. Here are our responses to your comments.
>
> [Q1: Relation between Equations 1 \& 3]
> Thanks for this suggestion. As optimizing Equation 1 is generally NP-hard, we opt for Equation 3 as an alternative solution.
>
> [Q2: Forward \& Backward Phases]
> This is indeed a very good point, which we initially did not try experimentally to avoid additional complexity. We created a variant, Forward-Backward, by equipping Causal Screening with the interleaving forward-backward selection strategy[1], which performs the forward phase and backward phase alternatively. We carried out additional experiments, and the table below summarizes the performance w.r.t. ACC-AUC and time cost per graph. Wherein, (s) and (c) denote the standard and cluster-based versions of our method, respectively. We found that:
> 1. Forward-Backward achieves lower ACC-AUC scores than Causal Screening. One possible reason is that, the forward phase measures the ICE of an edge conditioned on the previously-selected edges, while removing some edges the backward phase might distort the discovered sequence of edges. This again verifies the effectiveness of ICE estimation in Causal Screening.
> 2. Moreover, the forward-backward variant has higher computational costs than Causal Screening.
>
> |  	| Mutag. (s) 	| Mutag. (s) 	| REDDIT. (c) 	| REDDIT. (c) 	| Visual Genome (s) 	| Visual Genome (s) 	|
> |-	|-	|-	|-	|-	|-	|-	|
> |  	| ACC-AUC 	| Time 	| ACC-AUC 	| Time 	| ACC-AUC 	| Time 	|
> | Forward-Backward 	| 0.930 	| 2.34 	| 0.823 	| 5.18 	| 0.856 	| 0.443 	|
> | Causal Screening 	| 0.995 	| 1.83 	| 0.832 	| 2.52 	| 0.930 	| 0.326 	|
>
> [Q3: Causal Attributions (Equations 2 \& 4) vs Potential Outcome Estimation]
> Thanks. From the perspective of causality (as the causal graph in Figure 2(a) in the revision shows), mutual information $I$ is the variable in response to the graph variable $\mathcal{G}$ and the prediction variable $\hat{y}$, reflecting how much information about the prediction observation can be inferred from the graph observation. In this work, for an individual graph, a potential outcome is the outcome of mutual information under a potential treatment (i.e., assigning the graph variable with a certain subgraph). The causal effect of the treatment is the difference between the potential outcome if the graph receives the treatment and potential outcome if it is under control. In a nutshell, the causal attribution estimations in Equations 2 and 4 match the definition of causal effect [2,3], which are correct.
>
> It is worthwhile highlighting that:
> 1. As different graphs instances have various sizes of edges, one edge of a graph hardly corresponds to the edge of another graph. This makes the treatment on a single edge unique for the graph, which further downgrades the average causal effect to individual causal effect;
> 2. Empirically, the potential outcome estimation is achieved by the approximation of mutual information --- the cross-entropy between the target prediction vector $\hat{y}$ and the interventional prediction vector over $S$ classes --- instead of a scalar prediction of a certain class. Moreover, the table below summarizes the performance comparison between these two estimations. Wherein, the 1st.Est. is based on the mutual information (our Causal Screening), while the 2nd.Est is based on the prediction scalar.
>
> |  	| Mutag.(s) 	| Mutag.(s) 	| Mutag.(s) 	| REDDIT.(c) 	| REDDIT.(c) 	| REDDIT.(c) 	| Visual Genome(s) 	| Visual Genome(s) 	| Visual Genome (s) 	|
> |-	|-	|-	|-	|-	|-	|-	|-	|-	|-	|
> |  	| ACC-AUC 	| CST 	| SC 	| ACC-AUC 	| CST 	| SC 	| ACC-AUC 	| CST 	| SC 	|
> | 2nd.Est.	| 0.918 	| 0.449 	| 0.377 	| 0.816 	| 0.083 	| 0.144 	| 0.819 	| 0.323 	| 0.493 	|
> | 1st.Est. 	| 0.995 	| 0.155 	| 0.278 	| 0.823 	| 0.187 	| 0.162 	| 0.930 	| 0.315 	| 0.316 	|
>
> As a result, the causal attributions in Equations 2 and 4 are able to measure the changes of prediction distribution as the ``goodness'' of interventions.
>
> [Q4: Efficacy with Varying Graph Size \& Density]
> Thanks. Actually, we have conducted experiments on three benchmark datasets, Mutagenicity, REDDIT-MULTI-5K, and Visual Genome, which vary in terms of domain, graph size, and density. The statistics of these datasets are summarized in Table 4 of Appendix A.2.1.
>
> Reference
> 1. Algorithms for large scale markov blanket discovery. FLAIRS'2003
> 2. Causality. 2009
> 3. Causal inference in statistics: A primer. John Wiley & Sons, 2016.

---

### Official Review · AnonReviewer3 · 2020-11-02
**Good paper with a different and useful approach to the explanation of graph networks but lacks clarity on various occasions**

**Rating:** 7
**Confidence:** 4

**Review:**

**Summary**
The paper proposes a procedure for identifying a subgraph $\mathcal{G}_K$ of a given size $K$ (measured by the number of edges) whose output through the GNN function $f$ is as close as possible to that of the full graph $\mathcal{G}$. The proposed method is a greedy approach which starts from an empty graph and gradually adds the next edge by minimizing the difference between the outputs using mutual information. Furthermore, to reduce the computational complexity, a node clustering is done on the graph and the attribution is applied first on the edges between $C$ identified clusters and then transferred to all edges.

**Quality**
The experiments are relatively thorough and the writing is fine.

**Clarity**
The description of the method starts clearly at the high level but then lacks important information when the actual approach is presented in detail towards the end. In addition, more  discussions, motivations, and intuitions are needed for the design choices of the actual algorithm. Also, it is unclear if the main results are obtained using the cluster-based approach or not. Details described below.

**Originality**
While similar methods have been used in the image domain, to the best of the reviewer’s knowledge, the work is original in the context of graph networks. The reviewer believes the approach is better suited for graph networks, due to the feasibility arising from the structuredness of graphs.

**Significance**
The work conducts experiments covering different kinds of datasets as well as graph networks and considers multiple relevant baselines for GNN explanation. The results are, in general, significantly better than the baseline approaches on 3 different criteria. The paper is significant in that the approach is fundamentally different from the baselines and that the results are qualitatively and quantitatively different or better.

**Major technical comments**

*Experiments*
1. the results are obtained on three different datasets based on three different state-of-the-art graph networks and using an average of 5 independent runs.
2. Various graph explanation baselines are considered and the results show clear general improvement over the baselines in the three criteria used by the paper.
3. While the approach can generally be expensive, the time complexity of the cluster-based approach seems to be comparable to GNNexplainer and CXplain.
4. The difference in faithfulness (accuracy) compared to prior works on explaining graph predictions is clear and significant. However, the mere improvement over the baselines is not surprising since the proposed explanation method directly optimizes an objective to mimic the full graph’s function.
5. How is $\mathbf{X}$ obtained in practice? What clustering algorithm has been used for the method?
6. Does table 1 contain the results of the standard causal screening or the cluster-based one? The paragraph on the “influence of cluster numbers” seems to suggest the results in Table 1 belongs to the standard causal screening. If so, what are the results of the cluster-based method for w.r.t. contrastivity and sanity check? What is the computational complexity of the standard vs. cluster-based variant? What number of clusters is used for table 1, if it is cluster-based.

*Theory*
1. The greedy approach selects only *one possible* subgraph that explains the prediction. The distinction between “irrelevant edges” and “redundant edges” should be important here if the goal is to find the “causal” subgraph. Since the paper motivates the approach completely based on causal reasoning, an appropriate discussion is required.
2. How are the super-edges of the supernodes constructed? Does any edge between any nodes of two supernodes induce a single superedge between the two supernodes? Or there can be multiple super-edges connecting the two supernodes?
3. When doing the attribution on the new “super” graphs, are the intra-supernode edges always present when finding the attribution of superedges? How does this choice affect the goal of finding causal edges? Especially this choice virtually implying that intra-supernode edges are “attributed” when deriving the attribution of super-edges.
4. When constructing Z, why is the original X that contains soft clustering used while the H is obtained using the hard clustering assignments?
5. What is an interpretation of a z representation of a node? Why should the dot product of z vectors be a good proxy for their attribution? Separate arguments are probably needed for intra-supernode and inter-supernode edges.

**Minor technical comments**
+ the qualitative figures are interesting in that it shows the potential ways that the proposed approach can better find the key connections compared to the baselines.
- It’s better to start the “Task Description” section by something similar to “we define a graph of interest …”.  Otherwise it causes confusion since the definition, only based on edges, is in contrast to the previous paragraph and is only resolved at the end of the current paragraph.
- The statement “Directly optimizing Equation 1 is generally NP-hard” requires citation and/or explanation.
- in eq.3, shouldn’t it read: $|\mathcal{G}_k| = k$?

**Overall**
The paper proposes a different approach compared to the currently-existing explanations for graph networks. This type of “causal” explanation is considerably more feasible for and better fits graph networks thanks to their sparse and modularized structure. Although the results on accuracy are not surprising, they are still useful for applications that are interested in finding substructures responsible for a certain prediction. This is in fact quite commonly useful in many applications dealing with graph networks such chemistry, biology, image understanding, social networks, etc. The paper can greatly be improved on its clarity, especially when it comes to the design choices for the cluster-based variant. That being said, the results are quite convincing such that it empirically validates the approach.
All in all, if the paper is improved on clarity during the revision period, I would lean towards acceptance.

**Post-Rebuttal**
My main concern was regarding the clarity of the paper. I believe it is improved in the revised version, so I increase my rating.

---

> ### Author Response · Authors · 2020-11-19
> **Response to review #3 (part 1 of 2)**
>
> We greatly appreciate the reviewer's effort. Please find our responses to your questions and comments below:
>
> [Q1: Clarity of Cluster-based Causal Screening]
> Thanks for your valuable feedback. We have rephrased Section 3.3 in the revision. We also provide some points to answer your questions:
> - [How to obtain $\mathbf{X}$?] We used DiffPool[1], the differentiable clustering method.
> - [How are the super-edges of super-nodes constructed?] By assigning every node to the cluster with the largest confidence, we are able to simplify the graph as a new super-graph --- nodes in the same cluster are combined as a super-node; as such, intra-cluster edges are grouped as a self-loop of the super-node, while we integrate the parallel inter-cluster edges together as a super-edge across two different super-nodes. Here we treat these two types of super-edges equally, without distinguishing them.
> - [How to deal with intra-supernode edges?] We treat inter- and intra-cluster super-edges equally. Intra-cluster super-edges are not always present, once it is selected. Selecting one (inter- or intra-cluster) super-edge will validate all its component edges, and we feed it with the previous selection into the GNN model.
> - [Why $\mathbf{X}$ is soft, while $\mathbf{H}$ is hard?] We set $\mathbf{H}$ as hard, because one edge belongs to one and the only one super-edge, we could eliminate its inﬂuence on the CE estimations of other super-edges. We set $\mathbf{X}$ as a soft cluster assignment since one node would contribute to different clusters through its connected edges.
> - [Interpretation of $\mathbf{Z}$] We get inspiration from the link prediction task[9], which exploits the function of two endpoints as a proxy to approximate the ICE of an edge. Technically, we ﬁrst create the node representations via $\mathbf{Z} = \mathbf{X}\mathbf{H}$, where the v-th row $\mathbf{z}_v$ is the representation of node v that describes the contribution distributions over different clusters. Then, for an edge $e = (v, v ′ )$, we simply apply the inner product on the representations of endpoints $v$ and $v ′$ to approximate $e$’s attribution. Intuitively, if two endpoints have similar contribution distributions over clusters, the edge is highly likely to be inﬂuential to the information ﬂow along with the graph structure and contribute more to the GNN prediction. Through this way, we are able to estimate the ICEs of single edges and retrieve the top-K edges as the explanatory subgraph.
>
> [Q2: Clarity of Feature Types]
> Thanks. We discussed the differences among strongly relevant, weakly relevant (redundant), and irrelevant edges at the end of Section 3.2. We have improved our exposition in the revision. Following the literature of feature selection [5,6] and causality [7,8], we categorize edges into three groups based on the causal relationships with the predictions:
> 1. Strongly relevant edges contain unique information pertinent to GNN predictions, which are associated with significant causal contributions;
> 2. Weakly relevant edges are informative but redundant, which can be inferred and replaced by the previously selected edges. Hence, these edges have varnishing (close-to-zero) causal contributions to the GNN predictions;
> 3. Irrelevant edges do not bring any relevant information about predictions, whose causal contributions are zero. In a nutshell, the strongly relevant edges are more convincing and explainable than the redundant and irrelevant edges.
>
> Hence, the core of causal interpretability is to distinguish causal features (i.e., strongly relevant edges) from non-causal features (i.e., redundant edges and irrelevant edges).
>
> [Q3: Clarity of Task Description]
> Thanks for this suggestion. We have updated the ``Task Description'' section in the revision.
>
> [Q4: Equation 1 is NP-hard]
> Thanks for your suggestion. In the revised manuscript, we have added references to explain why the optimization of Equation 1 is NP-hard --- Directly optimizing Equation1 is generally NP-hard, largely due to the combinatorial nature of its constraint $|\mathcal{G}_{K}|=K$ with the number of possible subgraphs increasing sup-exponentially in the number of graph edges [2,3,4].
>
> [Q5: Constraint of Equation 3]
> Thanks. We used $|\mathcal{G}_{K}|=K$ in Equation 3 to indicate the total size of the explanatory subgraph. To improve the clarity of this equation, we have updated the constraint as $k=1,2,\cdots,K$.

---

> > ### Author Response · Authors · 2020-11-19
> > **Response to review #3 (part 2 of 2) due to the space limitation**
> >
> >
> > [Q6: Experimental Results of Standard vs Cluster-based Versions]
> > In the previous manuscript, we provided the implemental details in Appendix A.2.2 --- Based on different graph sizes, we applied the standard Causal Screening on the Mutagenicity and Visual Genome datasets, while employing the cluster-based version on REDDIT-MULTI-5K, and the results are shown in Table 1.  Also, we have compared the standard and cluster-based versions and presented the performance changes of the cluster-based Causal Screening over the cluster numbers in Table 3.
> >
> > The table below shows the performance comparison between the standard and cluster-based versions w.r.t. ACC-AUC, contrastivity (CST), sanity check (SC), and compute time, in the REDDIT-MULTI-5K dataset.
> >
> > | #Clusters        |    ------   | 5         |  10       |  15        |  20       |  25       |  30       |
> > |:------------------:|:----------:|:--------:|:--------:|:----------:|:--------:|:--------:|:----------:|
> > | ACC-AUC         |  0.508 ,|  0.816,   |  0.825,  |  0.823,  |  0.811,  |  0.819,  |  0.815  |
> > | CST                   |  0.078 ,|  0.086, |  0.168,  |  0.187,  |  0.098,  |  0.062,  |  0.066 |
> > | SC                     |  0.082,|  0.162,  |  0.167,  |   0.162,  |  0.187,  |  0.175,  |  0.182 |
> > |Compute Time| 72.38 ,| 1.010, |  2.520,  |  6.501,   |  8.810,  |  12.33,  |  15.80 |
> >
> > Reference:
> > 1. Hierarchical graph representation learning with differentiable pooling. NeurIPS'2018
> > 2. Learning to explain: An information-theoretic perspective on model interpretation. ICML'2018
> > 3. Causal discovery with reinforcement learning. ICLR'2020
> > 4. Gnnexplainer: Generating explanations for graph neural networks. NeurIPS'2019
> > 5. An introduction to variable and feature selection. JMLR'2003
> > 6. Wrappers for feature subset selection. AI'1997
> > 7. Beyond prediction: Using big data for policy problems. Science'2017
> > 8. Local causal and markov blanket induction for causal discovery and feature selection for the classiﬁcation part. JMLR'2010
> > 9. Factorization meets the neighborhood: a multifaceted collaborative ﬁltering model. KDD'2008

---

### Author Response · Authors · 2020-11-19
**General response**

We would like to thank all four reviewers for their time to offer thoughtful insights, comments, and suggestions. We have replied to each reviewer's comments individually below. In summary, we have updated our manuscript with the following changes:
- We have improved the clarity of our cluster-based Causal Screening (R3).
- We have improved the clarity of our exposition and writing of running example (R2), task description (R3), feature types (R3), and equations (R3, R1).
- We have added additional experimental results with (i) showcases of the causal attributions of selected edges (R4), (ii) a random baseline (R2), (iii) a variant of Causal Screening with a backward phase (R1), and (iv) performance comparison between the standard and cluster-based versions (R3).

These changes have increased the length of the main part of the paper to 9 pages. We believe that this is justified for improved clarity.

---

### Decision · Program_Chairs · 2021-01-07
**Final Decision**

**Decision:**

Reject

**Comment:**

I found the main algorithmic contributions to be interesting and of potential value to practitioners, as highlighted by Reviewer 3. Like several reviewers, I found the causal framing to be confusing, or at least not really to be framed in a framework like Pearl's: the word "confounding" is thrown a few times in the manuscript, but there is no formal sense by which it is linked to what we commonly understand as confounding. We are still talking about what happens inside a predictive model (a deterministic function), not what happens in the real world (the authors are not alone as targets of my observation: my point applies to a lot of the papers in the references, where the causal interpretation is hardly illuminating for those coming from a causal inference background, for instance). The reply to Reviewer 2, for instance, cites [1], which is about Granger causality and has little to do with Pearl's framework. Despite its name, Granger "causality" is a probabilistic concept (or, at best, an idea for identifying non-causality) with a very minimal causal basis besides the use of time ordering. A much more rigorous explanation of confounding in this paper's context needs to be provided.

That been said: as helpfully highlighted by Reviewer 3 (and summarized without any need to resort to a causal framing), there are several positive contributions added here, which might be of interest to the ICLR audience. The causal framing unfortunately gets in the way without adding clarity.

In its present state, the paper is not yet ready for publication. We hope that the reviewer comments prove helpful for preparing a strong future submission.